# The Accuracy and Absolute Reliability of a Knee Surgery Assistance System Based on ArUco-Type Sensors

**DOI:** 10.3390/s23198091

**Published:** 2023-09-26

**Authors:** Vicente J. León-Muñoz, Fernando Santonja-Medina, Francisco Lajara-Marco, Alonso J. Lisón-Almagro, Jesús Jiménez-Olivares, Carmelo Marín-Martínez, Salvador Amor-Jiménez, Elena Galián-Muñoz, Mirian López-López, Joaquín Moya-Angeler

**Affiliations:** 1Department of Orthopaedic Surgery and Traumatology, Hospital General Universitario Reina Sofía, 30003 Murcia, Spain; drlajaramarco@gmail.com (F.L.-M.); drajla78@gmail.com (A.J.L.-A.); camarin22@gmail.com (C.M.-M.); salva_am7@hotmail.com (S.A.-J.); elenagalianm@gmail.com (E.G.-M.); jmoyaangeler@gmail.com (J.M.-A.); 2Instituto de Cirugía Avanzada de la Rodilla (ICAR), 30005 Murcia, Spain; 3Department of Orthopaedic Surgery and Traumatology, Hospital Clínico Universitario Virgen de la Arrixaca, 30120 Murcia, Spain; fernando@santonjatrauma.es; 4Department of Surgery, Paediatrics and Obstetrics & Gynaecology, Faculty of Medicine, University of Murcia, 30120 Murcia, Spain; 5Department of Orthopaedic Surgery and Traumatology, Hospital Vega Baja, 03314 Orihuela, Spain; jesusjimenezolivares@gmail.com; 6Department of Information Technologies, Subdirección General de Tecnologías de la Información, Servicio Murciano de Salud, 30100 Murcia, Spain; mirindalopez@gmail.com

**Keywords:** accuracy, distance, angle, augmented reality (AR), orthopaedics, knee

## Abstract

Recent advances allow the use of Augmented Reality (AR) for many medical procedures. AR via optical navigators to aid various knee surgery techniques (e.g., femoral and tibial osteotomies, ligament reconstructions or menisci transplants) is becoming increasingly frequent. Accuracy in these procedures is essential, but evaluations of this technology still need to be made. Our study aimed to evaluate the system’s accuracy using an in vitro protocol. We hypothesised that the system’s accuracy was equal to or less than 1 mm and 1° for distance and angular measurements, respectively. Our research was an in vitro laboratory with a 316 L steel model. Absolute reliability was assessed according to the Hopkins criteria by seven independent evaluators. Each observer measured the thirty palpation points and the trademarks to acquire direct angular measurements on three occasions separated by at least two weeks. The system’s accuracy in assessing distances had a mean error of 1.203 mm and an uncertainty of 2.062, and for the angular values, a mean error of 0.778° and an uncertainty of 1.438. The intraclass correlation coefficient was for all intra-observer and inter-observers, almost perfect or perfect. The mean error for the distance’s determination was statistically larger than 1 mm (1.203 mm) but with a trivial effect size. The mean error assessing angular values was statistically less than 1°. Our results are similar to those published by other authors in accuracy analyses of AR systems.

## 1. Introduction

Using manual measuring instruments, such as graduated rulers, callipers, and goniometers, to determine distances and angular values is frequent in knee surgery. Any valid measuring device must be precise, accurate, and show adequate resolution and sensitivity. To be more accurate in our surgeries, we have progressively introduced highly accurate spatial measurement systems in operating rooms (OR). Since the late 1990s, computer-assisted surgery (CAS) has been introduced as an aid for knee surgery, mainly in prosthetic surgery, with the primary objective of increasing geometric precision [1,2]. In addition, CAS can provide the surgeon with real-time and 3D information [3]. CAS systems operate based on a virtual element of the natural anatomical one. This virtual element is extracted by prior digitalisation of studies by the image of the anatomical one or by intraoperative digitalisation, based on the acquisition of reference points that are processed to establish the geometry of this anatomical structure. The navigator consists of a computer platform, a tracking system, and a series of markers. The computing platform coordinates the flow of information, interprets it, and automatically performs the relevant mathematical and logical operations according to a given sequence of instructions. The tracking system is the communication mechanism between the markers on the anatomical element and the computer platform. Due to their reliability and accuracy, the most used tracking systems are optical trackers using infrared electromagnetic radiation (active or passive reflecting diodes). Several infrared detection cameras record the exact position of the emitters in an orthogonal coordinate system. With this type of system, a mean error of less than 1 mm or 1° (*p* < 0.001) has been estimated [2]. A limitation of the navigation systems is that they force surgeons to look away from the surgical field and verify the navigated surgical gestures on a flat-screen monitor [4].

Another drawback of conventional CAS systems is that the representation of the surgical gesture will occur in a completely virtual world, demanding the surgeon’s spatial-visual and oculomotor coordination skills to translate this virtual world into the real physical world. In addition, the surgeon sees a photorealistic simulation of the 3D model in a two-dimensional image on the monitor. Recent advances allow the use of Augmented Reality (AR) for many medical procedures. Milgram and Kishino described in 1994 the overlap between the physical and digital worlds and placed AR in this reality–virtuality continuum [5]. AR assigns the interaction between virtual environments and the physical world, allowing both to intermingle (AR superimposes information on real objects in real-time) through a technological device (usually smart glasses). AR has the potential to circumvent the limitations of current navigation systems by improving anatomical visualisation, surgeon ergonomics and intraoperative workflow [4,6]. We have developed some experience in different AR-assisted knee surgery techniques using optical surgical navigation with ArUco-type artificial marker sensors [7,8,9]. ArUco is a minimal library for AR applications based exclusively on OpenCV (Open Source Computer Vision Library) that relies on black-and-white (b/w) markers with codes detected by calling a single function [6].

A growing number of publications report clinical applications of AR in orthopaedic surgery [6,10,11,12,13,14]. Concerning knee surgery, these publications address prosthetic surgery, assistance in arthroscopic techniques (e.g., ligament reconstruction surgery), remote surgical assistance, immersive technology-based learning processes, and assistance in rehabilitation processes and gait analysis studies. This is, therefore, a broad clinical area of application for AR. Despite the increasing communication of clinical applications of such extended realities systems, this technology is indeed evolving. In our department, we have evaluated the spatial positioning system using ArUco markers detected by optical cameras on cadaveric specimens and in actual surgery (following regulatory approval for such use). The first tests were performed on cadaveric specimens just over three years ago. We have used the system in high tibial osteotomies, valgus and varus femoral osteotomies, derotational femoral osteotomies, anterior cruciate ligament reconstructions and menisci transplants.

Our study aimed to evaluate the accuracy of measurements performed with ArUco marker pointers detected by optical cameras using an in vitro protocol, adding errors attributable to the observer. We hypothesised that the system’s accuracy was like conventional CAS systems and, therefore, equal to or less than 1 mm and 1° for distance and angular measurements, respectively.

## 2. Materials and Methods

Our research consists of an in vitro laboratory study to determine the accuracy of a navigation system based on detecting non-natural markers by optical cameras. It was decided to conduct an in vitro study because of the advantages of tight control of the physical environment, reduced cost, reduction of statistical errors and higher throughput. Due to its characteristics, the study does not require ethical approval.

### 2.1. In Vitro Experimental Model

Together with the bioengineers at PQx Planificación Quirúrgica (PQx Planificación Quirúrgica, Murcia, Spain), a MedTech Start-Up with which we plan some of our interventions, we developed a model that would allow us to accurately locate several points (simulating the palpation of bony landmarks during knee surgery). The model was manufactured from 316 L steel, according to AISI (American Iron and Steel Institute) standards (Cr (16.5/18.5%) Ni (10.5/13.5%) Mo (2/2.25%) C (<0.03%)). It is an austenitic stainless steel, which is neither magnetic nor hardenable. It stands out from other steels due to the presence of 2–2.5% molybdenum, which provides it with a high corrosion resistance. We manufacture the feeler stylus from 17-4PH—steel (AISI 630) ((Cr (15/17%) Ni (3/5%) Cu (3/5%) Nb 5xC/0.45% C (<0.07%)). AISI 630 is a martensitic hardenable, stainless steel with high wear resistance, good corrosion resistance and high yield strength.

The model was designed with TopSolid’Design software V7.15 (TopSolid, Missler Software, Évry, France). TopSolid is an integrated CAD/CAM software for designing and creating fully functional 3D parts. As shown in Figure 1, Figure 2 and Figure 3, a platform and towers were designed with thirty palpation points to simulate bony landmarks and with various trademarks to acquire direct angular measurements. Before each acquisition, the observer calibrated the system by palpating the (x, y, z) point of the model and three reference points. The distances and angles determined were relative to this (x, y, z) point.

### 2.2. ArUco Fiducial Markers

Fiducial markers are artificial landmarks added to a scene that help find point correspondences between images or between images and known models [9]. We employed ArUco-type artificial marker sensors for optical surgical navigation, a simple library for augmented reality applications that only use OpenCV and depend on b/w markers with codes that can be detected by calling a single function [7,15]. OpenCV is a natively built C++ software library for computer vision and machine learning. It is an Apache 2 licensed product. OpenCV was developed to facilitate artificial perception and provide a common architecture for computer vision applications.

The ArUco markers were produced using an Ultimaker S3 printer (3D printing system) with double extrusion and a 230 × 190 × 200 mm print volume (Ultimaker BV, Utrecht, The Netherlands). Polylactic acid (PLA) filament from Ultimaker (Ultimaker BV, Geldermalsen, The Netherlands) was employed for the print [9].

### 2.3. Software

Python 3.8.3 (Python Software Foundation, Wilmington, NC, USA) was used to create the marker detection software, and OpenCV 4.0.1 served as the computer vision library. For the graphics engine, Unity 2019.2.17f1 (Unity Software Inc., San Francisco, CA, USA) was utilised. The method used to calculate distances was the one proposed by Vector3.Distance: the system receives two points, a and b, in three dimensions and performs the magnitude operation (a − b), the magnitude operation being √((ax − bx)^2^ + (ay − by)^2^ + (az − bz)^2^) [9]. The angular calculation method proposed by Vector3.Angle was used: the system receives two three-dimensional vectors representing the two directions whose degree difference is to be found. These vectors are normalized, and then the scalar product is performed; the result is restricted between −1 and 1, its arccosine is obtained, and it is multiplied by 57.29578 (180/pi) to transform radians into degrees, returning the result in this magnitude [9].

### 2.4. Optical Sensing Elements

An OAK-D camera (A00110-INTL) from Luxonis Holding Corporation in Denver, (Denver, CO, USA), was employed for our study. With a Display Field of View (dFoV) of 81 degrees and a resolution of 12 MP (4032 × 3040), the OAK-D baseboard features three integrated cameras that implement stereo and RGB vision. These cameras are connected directly to the OAK system in modules for depth processing and artificial intelligence.

The detection of ArUco markers, like other types of fiducial markers, is affected by noise, blur and occlusion despite relative immunity to light variations [9,16]. To avoid bias, we maintained stable conditions for the camera, monitor and model position throughout the investigation, with a blue background using surgical drapes and observers wearing blue surgical gowns of the same blue colour. OR lamps generate an illuminance over the surgical field of between 10,000 and 100,000 lux, which is excessive for our optical cameras. In the OR, it is also recommended to set a minimum of 2000 lux around the surgical table and 1000 lux in the whole room. We maintained a stable illuminance of 300 lux for our test in an 80 m^2^ experimental room equipped with light-emitting diode (LED) technology.

### 2.5. Absolute Reliability and Observers

Absolute reliability was evaluated according to the Hopkins criteria (minimum *n* of 30, at least six blinded assessors, at least three tests per observer, separated by at least two weeks) [17,18]. We carried out the study with seven independent evaluators with different experience levels. The following variables were noted as being specific to each observer: age, dominance, years of orthopaedic surgery practice, experience in navigation systems in orthopaedic surgery, experience in arthroscopic surgery, regular gaming with video consoles or leisure use of virtual or augmented reality systems. According to research by several authors, regular use of gaming devices or involvement in sports demanding substantial hand–eye coordination has a good effect on how well surgical skills like arthroscopic methods are learned [19,20]. We asked the observers about the usual play with two-dimensional (2D) and three-dimensional (3D) devices. Each observer measured the thirty palpation points and the twelve trademarks to acquire direct angular measurements on three occasions separated by at least two weeks. In each measurement session, each observer made three acquisitions.

### 2.6. Statistical Analysis

We performed statistical analysis using the Statistical Package for the Social Sciences (SPSS), version 25 for Windows (SPSS, Inc., Chicago, IL, USA). The average of the 1890 items for distance measurements and 756 items for angular values was used (each of the seven observers made 90 distance and 36 angle measurements in each of the three sessions separated by at least one week [270 length and 108 angle measurements per observer]) to evaluate the system’s validity. The Shapiro–Wilk test was used to check that the *p* values of the data were above the significance level of 0.05, with the null hypothesis that the data fit a normal distribution being accepted. All distributions met the normality criterion of this test. We use as reference values the distances between the (x, y, z) point, the different palpation points, and the different angular values defined in the design of the model using the CAD/CAM software. We used an arithmetic methodology that Lustig et al. [2] employed to establish comparisons between the accuracy of conventional CAS systems and non-natural fiducial mark detection systems. For each distance between the (x, y, z) point and the 30 points of the in vitro model, we calculated the difference between the authentic distance (D) and the distance sensed by the observer (D’). We considered this difference to be the error x during the acquisition of distances with the optical navigation system. The criteria used to define the method’s accuracy were the mean error x (in mm) and the corresponding uncertainty U = 2 × σ (σ being the standard deviation). The 95% confidence interval of the mean error was defined as (x − U; x + U). For each angular value, the difference between the real angular value (A) and the value sensed by the observer (A’) was calculated. This difference was considered to be the error x during the acquisition of angles with the optical navigation system. The criteria used to define the method’s accuracy were the mean error x (in degrees) and the corresponding uncertainty U = 2 × σ (σ being the standard deviation). The 95% confidence interval of the mean error was defined as (x − U; x + U). We also calculated the validity or degree of agreement between the mean value obtained from a large set of measurements and the actual value (MBE, mean bias error), the reliability (SD), the Standard Error of the Sample (SEM), and the intraclass correlation coefficient of absolute concordance using a two-factor random effects model [ICC (2,1)] [21]. Intra- and inter-observer reliability according to the criteria by Landis and Koch (<0 indicates no agreement, 0.00 to 0.20 indicates slight agreement, 0.21 to 0.40 indicates fair agreement, 0.41 to 0.60 indicates moderate agreement, 0.61 to 0.80 indicates substantial agreement, and 0.81 to 1.0 indicates almost perfect or perfect agreement) were calculated [22]. We calculated as well RMSE (Root Mean Square Error or Root Mean Square Deviation), MAE (Mean Absolute Error), and Mean Squared Error (MSE). In addition, we quantified the effect size using Cohen’s d-value (the difference between means) and the correlation coefficient (r), i.e., the magnitude of the association. The d-value to quantify the magnitude of an effect (the difference between means) can be interpreted according to the criteria by Hopkins et al. [23]: less than 0.2, trivial; 0.2 to 0.59, small; 0.6 to 1.19, moderate; 1.20 to 2, large; 2.1 to 3.99, very large, and greater than 4, extremely large. For the correlation coefficient, Cohen considers that a large effect corresponds to r = 0.5, medium to r = 0.3, and small to r = 0.1 [24].

## 3. Results

### 3.1. Accuracy of Distance Measurement

The results are shown in Table 1. The accuracy of the navigation system based on detecting non-natural markers by optical cameras to assess distances had a mean error of 1.203 mm and a maximum error of 6.7 mm. The uncertainty was 2.062 (95% confidence interval of the mean error −0.860–3.266). The mean error was statistically larger than 1 mm (*p* < 0.001). The effect size using Cohen’s d value was 0.003 (trivial), and the correlation coefficient r was 0.002 (small correlation).

### 3.2. Accuracy of Angle Measurement

The results are shown in Table 1. The accuracy of the navigation system based on detecting non-natural markers by optical cameras to assess angular values had a mean error of 0.778° and a maximum error of 4.43°. The uncertainty was 1.438 (95% confidence interval of the mean error −0.660–2.216). The mean error was less than 1° (*p* < 0.001). The effect size using Cohen’s d value was 0.034 (trivial), and the correlation coefficient r was 0.017 (small correlation).

A 6.88% (130/1890) of the considered errors in distance measurements (error measure ≠ 0.0 mm) and 4.1% (31/756) of the considered errors in angular measurements (error measure ≠ 0.0°) fall outside the mean value + 2SD.

The percentage distribution of distance and angle measurements are shown in Table 2.

In the intra-observer and inter-observer sample reliability analysis (intraclass correlation coefficient of absolute concordance using a two-factor random effects model [ICC (2,1)]), we obtained almost perfect or perfect agreement in all tests, according to Landis and Koch criteria [22]. The intra-observer results are shown in Table 3.

We did not observe any significant correlation between the errors in the determination of distance and angular values and observer-specific variables (age, dominance, years of orthopaedic surgery practice, experience in navigation systems in orthopaedic surgery, experience in arthroscopic surgery, regular gaming with video consoles or leisure use of virtual or augmented reality systems).

The average time taken for all acquisitions was 197.86 s, with a standard deviation of 29.691 s (range 154–291 s). Figure 4 shows the simple dispersion of time spent on acquisitions per observer per day. Less experienced observers who routinely play with video console devices or use virtual or augmented reality systems for leisure activities with some frequency have significantly shorter point acquisition and angle determination times than more experienced observers used to navigation-assisted surgery and arthroscopic surgery. We observed a significant negative Pearson correlation (−0.052) (*p* = 0.024) between time spent on acquisitions and distance errors. However, there is no significant difference in the mean errors in the inter-observer distances. We observed no correlation between time and errors in angular determination.

## 4. Discussion

The main aim of our study was to determine the inaccuracy that occurs in the acquisition of points in space using an ArUco marker system detected by optical cameras in an in vitro protocol. Contrary to other studies [2], the system’s accuracy was analysed, and the effect of the surgeon was included, with seven observers with different levels of experience. According to our results, the distances evaluated showed a mean error of more than 1 mm (1.203 mm; *p* < 0.001), which is a trivial effect size using Cohen’s d value (0.003), and the angle of determination had a mean error of less than 1° (0.778°; *p* < 0.001). In addition, intra- and inter-observer variability was minimal. To the best of our knowledge, no studies have assessed this type of fiducial marker’s accuracy fulfilling absolute reliability criteria [17,18].

The use of AR in surgery, especially orthopaedic surgery, is not science fiction. It is a reality [9,25,26,27,28,29]. It is an aid to locate different elements (whether anatomical or instrumental) unambiguously in a Cartesian space by means of coordinates and in real time. Just as we use different measuring systems during surgery (e.g., calibrated ruler, goniometer or calliper) that are reliable because they have been validated, validation of the accuracy and reliability of these novel measuring instruments is necessary. The clinical application of our article was the validation of this particular system.

Can we accept a 6 mm or 4° error in knee surgery? No, obviously not. Our hypothesis to explain these maximum errors is that they are due to the operators. If there is a movement of the sensors just at the time of the measurement acquisition, the measurement will be in error. The significant negative Pearson correlation between the errors in determining the distances and the time spent in the recording would support this hypothesis. Therefore, to evaluate the actual accuracy of the system, it would be essential to experiment with conditions of absolute independence of human error, as has been done with other systems [2].

Technological innovation in surgical workflows is necessary, but in innovating, we are always at risk of a certain technological arrogance if we do not contribute to augmented humanity, as defined by Guerrero et al. [30] as a human–computer integration that expands the limits of human function by extending physical, intellectual and social capacities to increase performance and productivity. The discussion often focuses on balancing the clinical benefit of technological innovation with the added time and cost usually associated with it. One avoidance of such technological arrogance is to increase accuracy (or at least precision) relative to the pre-innovation standard.

Several authors have published trials evaluating the accuracy of extended reality (XR) systems concerning hip [31,32,33,34,35] and knee surgery [26,27,28,29].

Fallavollita et al. [36] evaluated whether an AR technology can provide accurate, efficient, and reliable intraoperative alignment control of the mechanical axis deviation (distance from the knee joint centre to the line connecting the hip and ankle joint centres). A camera-augmented mobile C-arm (CamC) was used as AR technology, and five participants with different surgical experience levels determined the anatomical landmarks defining the mechanical axis and centre of the knee in twenty-five human cadaveric specimens. The authors demonstrated that the AR technology provides accurate mechanical axis deviation (no significant difference between CamC and CT values and a strong positive correlation, which means that CamC values go with the CT values) [36].

Tsukada et al. [28] performed a pilot study to examine the accuracy of an imageless navigation system using AR technology with resin markers with a square two-dimensional bar code (the system allows the surgeon to view the tibial axis superimposed on the surgical field through the display of a smartphone, similar to Ogawa et al. [35]) for total knee arthroplasty (TKA). Using the system, one surgeon resected ten pairs of tibial sawbones by viewing the tibial axis and aiming varus/valgus, posterior slope, internal/external rotation angles, and resection level superimposed on the surgical field. No significant differences existed between the angles displayed on the smartphone screen and the measurement angles determined using CT. The system indicated varus/valgus and posterior slope angles of less than 1° and internal/external rotation angles of less than 2° for the differences between the values displayed on the smartphone screen and the actual measurement values. Although the mean difference of 0.6 mm ± 0.7 mm in terms of the thickness of resected bone was also comparable to that of the conventional navigation systems, the authors [28] believe that the value of the mean difference can represent an unacceptable error for correct balancing during TKA.

Tsukada et al. [29] performed a two-phase study to evaluate the accuracy of the AR system in distal femoral resection during TKA. First, a total of ten femoral sawbone specimens were resected by a single surgeon. The absolute values of the differences between the angles measured using CT and the angles displayed on the smartphone screen were 0.8° ± 0.5° (range, 0.3° to 1.9°) in the coronal plane and 0.6° ± 0.5° (range, 0.0° to 1.4°) in the sagittal plane. Secondly, the authors performed a clinical study on 72 TKA (distal femoral resection with the AR system vs. with a conventional intramedullary guide, with the target of femoral coronal alignment perpendicular to the mechanical axis in both groups). In the clinical study (evaluating postoperative standing long-leg radiographs), the mean absolute value of the error in coronal alignment was significantly smaller in the AR-based navigation group than in the intramedullary guide group (1.1° ± 1.0° [range, 0.0° to 3.2°] compared with 2.2° ± 1.6° [range, 0.0° to 5.5°], respectively; 95% confidence interval, 0.5° to 1.8°; *p* < 0.001).

Recently, Tsukada et al. [27] evaluated the accuracy of the AR system (adapting Quick Response [QR] coded markers to the extramedullary cutting guide and pointer and with the system described in their previous publications [28,29], using a smartphone camera as a sensor) in tibial osteotomy of unicompartmental knee prostheses. The authors [27] published absolute differences between the preoperative target resection angles and postoperative measured angles of 1.9° ± 1.5° in the coronal alignment and 2.6° ± 1.2° in the sagittal alignment.

Iacono et al. [37] prospectively studied the alignment accuracy of the Knee+ AR navigation system in five consecutive patients (Pixee Medical Company, Besançon, France). The Knee+ system consists of smart glasses worn by the surgeon, a laptop, and specific QR-coded markers connected with tibial and femur resection guides. It allows the surgeon to view the tibial and femur axis superimposed on the surgical field through smart glasses. The authors [37] published a cutting error of less than 1° of difference in the femur and tibia coronal alignment and less than 2° about flexion/extension of the femur and posterior tibial slope. Bennett et al. [38] recently published a prospective, consecutive series of twenty patients undergoing TKA utilising the Knee+ system and determining the coronal and sagittal alignment of the femoral and tibial bone cuts measured on postoperative CT scans. The employed system produced a mean absolute error of 1.4°, 2.0°, 1.1° and 1.6° for the femoral coronal, femoral sagittal, tibial coronal and tibial sagittal alignments, respectively. The authors [38] concluded that AR navigation could achieve accurate alignment of TKA with a low rate of component malposition in the coronal plane but with more significant inaccuracy in the sagittal plane, which conditions some sagittal outliers.

Pokhrel et al. [26] propose a system based on a volume subtraction technique that considers the history of the area which has been cut and measures it for the target shape. The authors postulate that using fiducial markers and bulky trackers presents significant limitations in terms of errors and barriers to surgical setup. They propose adapting the 3D–2D shape-matching integration and stereoscopic tracking proposed by Wang et al. [39,40] into the Tracking Learning Detection algorithm for real-time registration with the Iterative Closest Point as the medium for overlay model refinement. Image registration tracks the patient and matches 3D contours in real-time without fiducial markers or references. The real-time autostereoscopic 3D images are implemented with the help of a Graphics Processing Unit. The remaining area to be cut is calculated with the volumetric analysis of the area already cut and the total area to be cut. With the proposed algorithm, the authors publish an increase in video accuracy to 0.40 to 0.55 mm overlay error [26].

Our results (mean error of 1.203 mm for the distances and 0.778° for the angles evaluation) are similar to those published by other authors in accuracy analyses of AR systems (mean minimum angular error values from 0.6° ± 0.5° up to mean maximum values of 2.6° ± 1.2°) [26,27,28,29] and values that do not differ significantly from those published for CAS-, PSI- or robotic-assisted knee surgery. The evaluated system may be suitable for knee surgeries, not limited to TKA surgery.

There are some limitations to our study. First, contrary to another study [2], we have not tested the technology’s accuracy in isolation by designing a system independent of the observer. Technology is not user-independent, so we decided to incorporate the possible effect of observer-induced error methodologically. Second, we designed similar conditions to the acquisition of anatomical knee landmarks (distances from 47.49 mm to 117.77 mm and angles from 35° to 81.04°), access difficulty levels for both right-handed and left-handed observers and boundary positions so that optical cameras captured the ArUco markers in conditions that were far from optimal and as “real” as possible; however, this is still an in vitro model that differs from the anatomy of a knee and from the texture of the knee tissues which are very different from 316 L Steel (as it is a model with well-defined indentations, we have avoided the bias due to observer variability in the determination of the anatomical landmarks). Keeping the conditions of the experimental room environment constant is also a strength, but it distances the research from the “real” conditions in the operating theatre. For example, we have determined the illuminance in our experimental room with an average value of 300 lux in 10 measurements at different points, which is lower than the usual illumination in an OR. This is a limitation under actual conditions but a strength in demonstrating that the optical detection of ArUco-type markers is accurate at low light. Third, we could have used smart glasses, but we decided that the entire trial would be conducted with direct vision of a monitor. We have yet to assess whether there is a difference in accuracy with the use of smart glasses, a study that may be of some interest. Fourth, this new and emerging technology is constantly evolving, so further studies and confirmation of the initial validation of our work will be required. Fifth, we have validated the accuracy and reliability of a specific system, so our results cannot be extrapolated to other AR systems, which will also require independent validation.

The strength of our research lies in the methodology, which was designed to meet the criteria for absolute reliability according to the Hopkins criteria [18]. In addition, we have included potential operator error to ensure that the conditions were similar to real-world conditions, as there is little point in demonstrating the extreme accuracy of a technology that is completely lost when used by the surgeon. Another strength is that, to the best of our knowledge, this is the first analysis of the accuracy and reliability of the adaptation of ArUco markers for use in orthopaedic surgery detected by optical cameras.

## 5. Conclusions

Based on absolute reliability criteria, the accuracy of measurements performed with ArUco marker pointers detected by optical cameras was evaluated using an in vitro protocol with the hypothesis that the system’s accuracy was equal to or less than 1 mm and 1° for distance and angular measurements, respectively. However, the mean error for the distance’s determination was statistically larger than 1 mm (1.203 mm) but with a trivial effect size. The mean error assessing angular values was statistically minor than 1°. Our results are similar to those published by other authors in accuracy analyses of AR systems. Although efforts must be made to optimise accuracy to the highest level, our study, which does not eliminate human error, indicates sufficient accuracy for applying this technology in knee surgery.

## Figures and Tables

**Figure 1 sensors-23-08091-f001:**
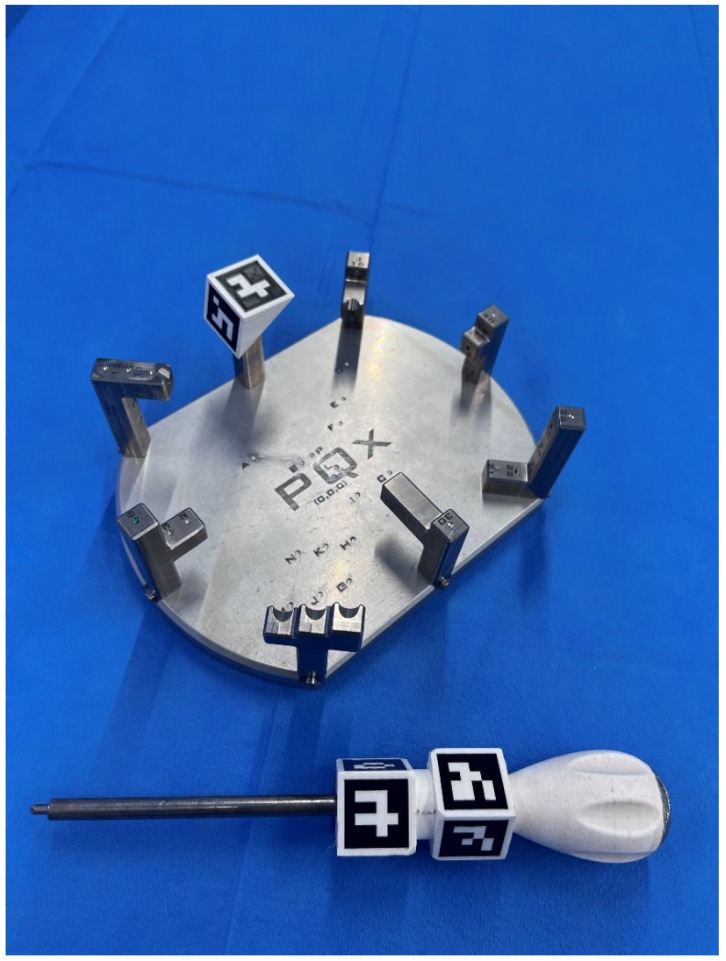
316 L steel model used for distance and angle measurements and stylus. Both have ArUco-type artificial markers. The relationship between the palpation points has been arranged with different levels of access difficulty for both right-handed and left-handed observers.

**Figure 2 sensors-23-08091-f002:**
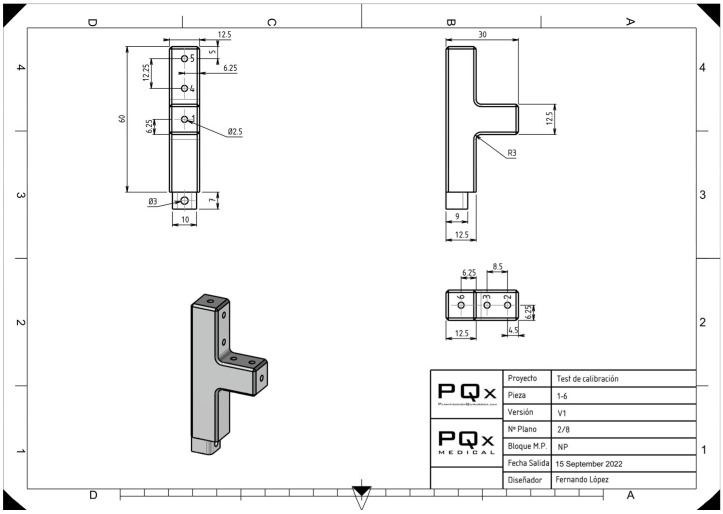
Design of one of the towers that make up the experimental model with the known references of the distances of each palpation point.

**Figure 3 sensors-23-08091-f003:**
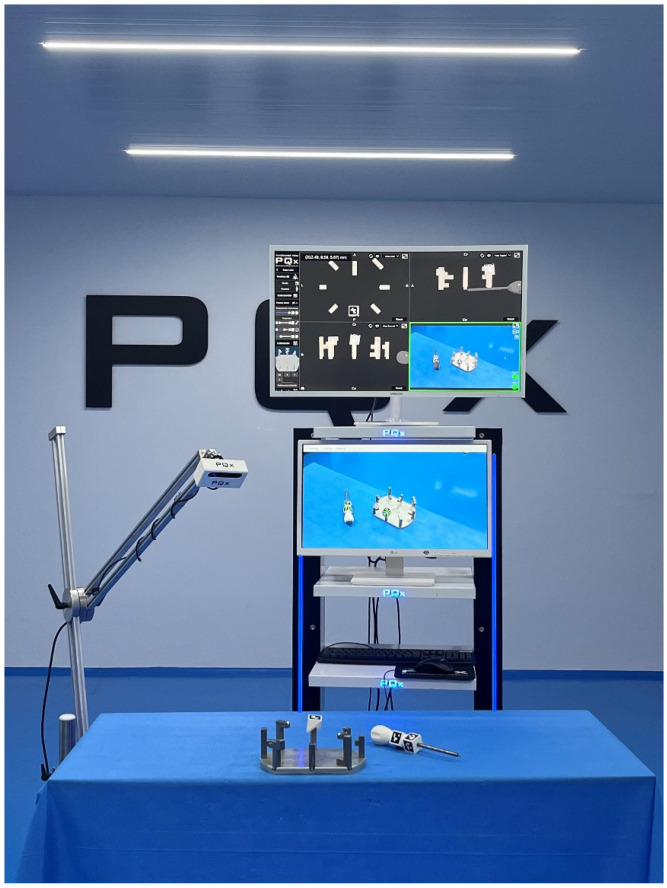
Devices used in the experiment: platform with palpation towers, optical camera, central processing unit and monitors. The monitors were positioned centrally in front of the observers so that the exchange of information between the real, virtual and AR domains could occur with a simple eye-raising gesture.

**Figure 4 sensors-23-08091-f004:**
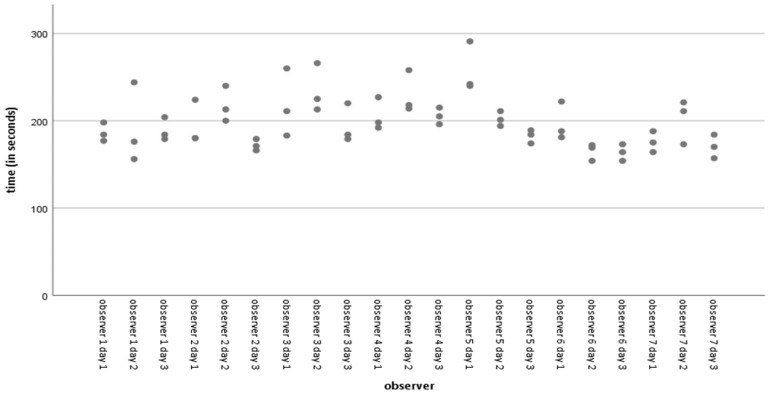
Simple dispersion with time line adjustment (in seconds) per observer.

**Table 1 sensors-23-08091-t001:** Distance and angular measurement mean errors with optical cameras detecting the ArUco-type artificial marker sensors.

	Distance Measurement	Angle Measurement
Number of acquisitions	*n* = 1890	*n* = 756
Mean error	1.203 mm	0.778°
Min	0.00 mm	0.00°
Max	6.70 mm	4.43°
Standard deviation	1.031 mm	0.719°
Uncertainty	2.062	1.438
Standard Error of the Sample	0.024 mm	0.026°
Root Mean Square Error	1.585 mm	1.495°
Mean Bias Error	0.051 mm	0.466°
Mean Absolute Error	1.203 mm	0.815°
Mean Squared Error	2.51 mm	2.22°

**Table 2 sensors-23-08091-t002:** Distribution of the measurements obtained.

	Distance Measurement	Angle Measurement
<1 mm or °	52.54% (993/1890) (89 measures < 0.1 mm)	70.24% (531/756) (71 measures < 0.1°)
between 1 and 2 mm or °	30.32% (573/1890)	25.26% (191/756)
>2 mm or °	17.14% (324/1890)	4.5% (34/756)

**Table 3 sensors-23-08091-t003:** Intra-observer reliability. The intraclass correlation coefficient of absolute concordance using a two-factor random effects model.

	Intraclass Correlation	95% Confidence Interval	
		Lower Bound	Upper Bound
Observer 1	0.999	0.998	0.999
Observer 2	0.995	0.992	0.997
Observer 3	0.999	0.998	0.999
Observer 4	0.999	0.999	1.000
Observer 5	0.995	0.992	0.997
Observer 6	0.999	0.998	0.999
Observer 7	0.999	0.998	0.999

## Data Availability

Not applicable.

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
