# Peer review of "The Accuracy and Absolute Reliability of a Knee Surgery Assistance System Based on ArUco-Type Sensors"

_sensors, 2023, doi:10.3390/s23198091_

Round 1

Reviewer 1 Report

the paper is dealing with an evolving topic in orthopaedic surgery and many surgeons would be interested in reading this

But the authors need to add a paragraph in the abstract and the conclusion about the clinical significance of the study and the findings. 

well written but need to address the interest of clinical community not just engineering 

Author Response

Dear reviewer 1,

Firstly, we would like to thank you for your comments and for allowing us to address the issues you raised in order to improve the quality of the manuscript. We greatly appreciate your observations and the time you have taken to provide constructive criticism and feedback on our manuscript.

We will respond to your comments and objections point by point and indicate the changes we propose to the manuscript to address your comments. We have highlighted the changes to the manuscript in red.

Comments and Suggestions for Authors

the paper is dealing with an evolving topic in orthopaedic surgery and many surgeons would be interested in reading this

But the authors need to add a paragraph in the abstract and the conclusion about the clinical significance of the study and the findings.

We have modified the abstract format to conform to the publisher's guidelines.

We added to the abstract: “AR via optical navigators to aid various knee surgery techniques (e.g., femoral, and tibial osteotomies, ligament reconstructions or menisci transplants) is becoming increasingly frequent. Accuracy in these procedures is essential, but evaluations of this technology still need to be made.”

We have added to the conclusions: “Although efforts must be made to optimise accuracy to the highest level, our study, which does not eliminate human error, indicates sufficient accuracy for applying this technology in knee surgery.”

well written but need to address the interest of clinical community not just engineering

I am terribly sorry that the article is so lacking in clinical style and is so biotech. I found it difficult to write because I am not a bioengineer; I am an orthopaedic surgeon, and I do knee surgery. However, we felt that the validation of the technology could only be done based on engineering assumptions. Once the system’s accuracy has been demonstrated, it will be time for articles on the clinical applicability of this type of technology. Both aspects together may be beyond the scope of a single manuscript. I hope you will understand this approach.

Reviewer 2 Report

I thank the editors for selecting me in the review process of this manuscript. The article is well written and I feel some minor corrections are needed which I list below: INTRODUCTION - Enrich by adding the following bibliographic reference: PMC9945668 MATERIALS AND METHODS - This section is a bit confusing. Reorganize the entire section by adding sub-paragraphs that can better orient the reader. Also add a sub-section about the statistical analysis used DISCUSSION - The authors in line 393 state that their results were similar to those obtained in studies 27-30. Describe what these studies found by comparing them with your results. - Add the strengths of your studio.

Author Response

Dear reviewer 2,

Firstly, we would like to thank you for your comments and for allowing us to address the issues you raised in order to improve the quality of the manuscript. We greatly appreciate your observations and the time you have taken to provide constructive criticism and feedback on our manuscript.

We will respond to your comments and objections point by point and indicate the changes we propose to the manuscript to address your comments. We have highlighted the changes to the manuscript in red.

I thank the editors for selecting me in the review process of this manuscript. The article is well written and I feel some minor corrections are needed which I list below:

INTRODUCTION - Enrich by adding the following bibliographic reference: PMC9945668

I have consulted the reference provided (Scaturro D, Vitagliani F, Caracappa D, Tomasello S, Chiaramonte R, Vecchio M, Camarda L, Mauro GL. Rehabilitation approach in robot assisted total knee arthroplasty: an observational study. BMC Musculoskelet Disord. 2023 Feb 22;24(1):140. doi: 10.1186/s12891-023-06230-2. PMID: 36814210; PMCID: PMC9945668). I have read the article and looked for the match in the introduction of our manuscript, but I cannot find the sentence(s) to which your suggestion may refer. Please confirm that this is the article, and can you tell us where to fit the reference?

MATERIALS AND METHODS - This section is a bit confusing. Reorganize the entire section by adding sub-paragraphs that can better orient the reader. Also add a sub-section about the statistical analysis used

We have reorganised the M&M section into subsections to avoid confusion and clarify the methodology. We have divided the M&M section into the following subsections:

2.1 In vitro experimental model

2.2 ArUco fiducial markers

2.3 Software

2.4 Optical sensing elements

2.5 Absolute reliability and observers, and

2.6 Statistical analysis

We have also rearranged a paragraph to improve the description of the experiment, following your suggestion.

DISCUSSION - The authors in line 393 state that their results were similar to those obtained in studies 27-30. Describe what these studies found by comparing them with your results. - Add the strengths of your studio.

The description of the studies mentioned and the error values obtained by the authors can be found in the previous paragraphs: Tsukada et al. [27] in lines 366 to 378 of the R1 version of the manuscript, Tsukada et al. [28] in lines 379 to 385 of the R1 version of the manuscript, Iacono et al. [29] in lines 386 to 392 of the R1 version of the manuscript and Bennett et al. [30] in lines 393 to 401 of the R1 version of the manuscript. We wanted to avoid repeating the results so as not to repeat information. We have added the following paragraph to contextualise our results with those of the other authors: “(mean minimum angular error values from 0.6° ± 0.5° up to mean maximum values of 2.6° ± 1.2°)”

We have added the following paragraph as a strength of our research: “The strength of our research lies in the methodology, which was designed to meet the criteria for absolute reliability according to the Hopkins criteria. In addition, we have included potential operator error to ensure that the conditions were similar to real-world conditions, as there is little point in demonstrating the extreme accuracy of a technology that is completely lost when used by the surgeon. Another strength is that, to the best of our knowledge, this is the first analysis of the accuracy and reliability of the adaptation of ArUco markers for use in orthopaedic surgery detected by optical cameras.”

Reviewer 3 Report

Comments to the manuscript ID sensors-2570573 entitled: The accuracy and absolute reliability of a knee surgery assistance system based on ArUco-type sensors. This is a great manuscript that explain the reliability of a new device for assisting in the knee surgery. The manuscript is well structure but some changes should be improved.

0.      Abstract

Is well structured and explained all the sections properly.

1.      Introduction.

Is an interesting introduction and it reflects the aim of the study and hypothesis well explained. Congratulations.

2.      Material and Methods.

The authors refers in all the manuscript as “we developed”, “we designed”…. Ej: line 101,102,103,112…. Authors must change these and put it in proper scientific manners.

Instead of that appreciation the material and methods are well explained.

3.      Results

Lines 239-259: The table does not reflect the data explained in the manuscript. Please rewrite the table and add all the data showed.

Lines 260-271: It should be interesting add the table about these data.

Lines 279-287: The same explanation with a table should be done.

4.      Discussion

Lines 307-312: Please explain better with your own words and do not copy-paste a cite from other researcher.

Changes the grammar mistakes and adapt to the scientific manners.

5.      Conclusions

Delete “we evaluated”.

Authors must adat the manuscript to a correct english grammar

Author Response

Dear reviewer 3,

Firstly, we would like to thank you for your comments and for allowing us to address the issues you raised in order to improve the quality of the manuscript. We greatly appreciate your observations and the time you have taken to provide constructive criticism and feedback on our manuscript. We sincerely thank you for your congratulations on certain sections of our manuscript.

We will respond to your comments and objections point by point and indicate the changes we propose to the manuscript to address your comments. We have highlighted the changes to the manuscript in red.

Comments to the manuscript ID sensors-2570573 entitled: The accuracy and absolute reliability of a knee surgery assistance system based on ArUco-type sensors. This is a great manuscript that explain the reliability of a new device for assisting in the knee surgery. The manuscript is well structured, but some changes should be improved.

  1. Abstract

Is well structured and explained all the sections properly.

  1. Introduction.

Is an interesting introduction and it reflects the aim of the study and hypothesis well explained. Congratulations.

  1. Material and Methods.

The authors refers in all the manuscript as “we developed”, “we designed”…. Ej: line 101,102,103,112…. Authors must change these and put it in proper scientific manners.

Thank you for pointing that out. In our native language, it is considered correct to use the passive voice. In English it is not so correct, and we have used this grammatical form to avoid the passive voice, but I agree with you that it is not appropriate, and we have changed this way of expressing ourselves throughout the manuscript.

Instead of that appreciation the material and methods are well explained.

At the request of another reviewer, we have divided M&M into subsections.

  1. Results

Lines 239-259: The table does not reflect the data explained in the manuscript. Please rewrite the table and add all the data showed.

We wanted to present the most relevant data in the table and avoid duplication in the text and table. For this reason, the information on the Standard Error of the Sample, the Root Mean Square Error, the Mean Bias Error, the Mean Absolute Error, the Mean Squared Error, the effect size using Cohen's d and the correlation coefficient r of the distance measurements, and the angle measurements are not shown in the table. If the reviewer prefers to present the data in the table, we suggest removing the less relevant data from the results from the text so as not to duplicate the information. If this option does not seem appropriate, please let us know so we can change it.

Lines 260-271: It should be interesting add the table about these data.

We add the information in tabular form and remove the presentation of the results as a paragraph from the text, following the same concept formulated above, to avoid duplication of information in the text and table. We emphasise that although we are in favour of avoiding this duplication, if the reviewer considers the presentation in text and table necessary, we can modify it if indicated.

Lines 279-287: The same explanation with a table should be done.

In the case of the presentation of the results relating to the time taken to acquire the measurements, we present the results of only two values, so it seems interesting to present the distribution of time per observer in the form of a graph. We insist that we have no objection to presenting the results in a table, but the results relating to the time taken to acquire the measurements require a table that is too large due to the number of variables.

  1. Discussion

Lines 307-312: Please explain better with your own words and do not copy-paste a cite from other researcher.

At your suggestion, we have changed the sentence to: “Technological innovation in surgical workflows is necessary, but in innovating, we are always at risk of a certain technological arrogance if we do not contribute to augmented humanity, as defined by Guerrero et al. [20] as a human-computer integration that expands the limits of human function by extending physical, intellectual and social capacities to increase performance and productivity.”

Changes the grammar mistakes and adapt to the scientific manners.

We have checked the grammatical construction of the entire manuscript.

  1. Conclusions

Delete “we evaluated”.

We have removed it.

Round 2

Reviewer 1 Report

the paper is interesting with the growing trend of use of VR in Orthopaedics 

The paper still present a very preliminary and evolving technology. 

I m not sure about the clinical significance for orthopaedic surgery doctors 

That need to be explained in the introduction and the discussion and the limitations and further research needed and future direction 

Author Response

We would like to thank reviewer 1 for his comments. We believe that a satisfactory response will improve the quality of the manuscript. We appreciate the time and effort put into proofreading the manuscript.

We will respond to your comments and objections point by point and indicate the changes we propose to the manuscript to address your comments. We have highlighted the changes to the manuscript in yellow.

the paper is interesting with the growing trend of use of VR in Orthopaedics

Thank you for your interest in the manuscript. We agree that extended realities (in particular augmented reality) are increasingly being incorporated as an assistive technology in various areas of orthopaedic surgery.

The paper still present a very preliminary and evolving technology.

This technology is indeed evolving. That is why our study, which attempts to validate the level of accuracy that can be achieved with this type of system today, makes sense.

I disagree with you that this is a preliminary technology. The system we have evaluated is commercially available for use, for example, to facilitate the performance of various orthopaedic procedures. I have personally tested the system, both on cadaveric specimens and in real surgery (following regulatory approval for such use). The first tests were performed on cadavers just over three years ago. We have used the system in high tibial osteotomies, femoral varus and valgus osteotomies, femoral derotational osteotomies, ACL reconstructions and menisci transplants. In all of these operations, we required the technology to provide spatial positioning at all times and a correlation between reality and the virtuality we saw on the screen. The manufacturer provided us with data on the accuracy of the system, but there was no rigorous independent analysis of accuracy. This, in our opinion, justified the study that we presented in our manuscript.

I m not sure about the clinical significance for orthopaedic surgery doctors

The clinical significance for orthopaedic surgeons is not so much the existence of the technology, but rather the validation of its accuracy through proper methodology. Just as we want to rely on the readings of a calibrated ruler, goniometer, or calliper, we want to rely on the readings of these new technologies. The only way to do this is to assess their accuracy and reliability.

That need to be explained in the introduction and the discussion and the limitations and further research needed and future direction

We have tried to emphasise these concepts, which I mention to you in the manuscript. I have also tried to make clear the limitations of our research, the limitations of the technology and the need for further studies in the future. We have highlighted the paragraphs in yellow in the R2 version of the manuscript. Thank you again for the suggested corrections and the focus on the clinical application of our research.